# Pharmaco-Toxicological Effects of Atypical Synthetic Cathinone Mephtetramine (MTTA) in Mice: Possible Reasons for Its Brief Appearance over NPSs Scene

**DOI:** 10.3390/brainsci13020161

**Published:** 2023-01-18

**Authors:** Giorgia Corli, Micaela Tirri, Raffaella Arfè, Beatrice Marchetti, Tatiana Bernardi, Martina Borsari, Sara Odoardi, Serena Mestria, Sabina Strano-Rossi, Margherita Neri, Rosa Maria Gaudio, Sabrine Bilel, Matteo Marti

**Affiliations:** 1Department of Translational Medicine, Section of Legal Medicine, LTTA Center and University Center of Gender Medicine, University of Ferrara, 44121 Ferrara, Italy; 2Department of Environmental Sciences and Prevention, University of Ferrara, 44121 Ferrara, Italy; 3Forensic Toxicology Laboratory, Department of Health Surveillance and Bioethics, Università Cattolica del Sacro Cuore F. Policlinico Gemelli IRCCS, 00169 Rome, Italy; 4Department of Medical Sciences, University of Ferrara, 44121 Ferrara, Italy; 5University Center for Studies on Gender Medicine, University of Ferrara, 44121 Ferrara, Italy; 6Collaborative Center for the Italian National Early Warning System, Department of Anti-Drug Policies, Presidency of the Council of Ministers, 00186 Rome, Italy

**Keywords:** MTTA, mephtetramine, Novel Psychoactive Substances, sensorimotor, behaviour, mice, blood and urine analysis, synthetic cathinones, histological changes

## Abstract

Over the last year, NPSs have been steadily on the rise in the illicit drug market. Among these, synthetic cathinones seem to become increasingly popular among young adults, mainly because of their ability to replicate the effects of traditional psychostimulant drugs, such as cocaine, MDMA and amphetamines. However, scarce data are available about the in vivo pharmaco-toxicology of these new substances. To this end, this study focused on evaluation of effects induced by repeated administration of mephtetramine (MTTA 0.1–30 mg/kg i.p.) in mice. This atypical cathinone highlighted a sensorial (inhibition of visual and acoustic reflexes) and transient physiological parameter (decrease in breath rate and temperature) change in mice. Regarding motor activity, both a dose-dependent increase (accelerod test) and biphasic effect (drag and mobility time test) have been shown. In addition, blood and urine samples have been analysed to enrich the experimental featuring of the present study with reference to evaluation of potential toxicity related to consumption of MTTA. The latter analysis has particularly revealed important changes in blood cells count and blood and urine physicochemical profile after repeated treatment with this atypical cathinone. Moreover, MTTA induced histological changes in heart, kidney and liver samples, emphasizing its potential toxicity.

## 1. Introduction

Since new psychoactive substances (NPSs) have surfaced in the worldwide illicit drug market, they have quickly established themselves as a public health emergency and become one of the major international concerns [1]. To date, over 1100 different NPSs have been reported to the UNODC Early Warning Advisory (EWA) by its member states [2]. Despite efforts to counter it, the European drug market has also demonstrated greater diffusion and availability of NPSs with a diverse range of increasing purity or potency [3]. Even during the difficult lockdown period due to the COVID-19 pandemic, the NPSs market remained remarkably resilient, with successful deliveries of purchased drugs on the Darknet [4]. Unexpectedly, 3,4-methylenedioxy-N-methylamphetamine (MDMA), lysergic acid diethylamide (LSD), main amphetamine-like psychostimulants, 2,5-dimethoxy-4-bromophenethylamine (2C-B) and cocaine were the main purchased drugs [4]. These substances belong to the so-called ‘club drugs’ or ‘party pills’, commonly taken by young adults (age 15–35) who regularly attend pubs, nightclubs, music festivals and rave parties [5,6]. Among these, synthetic cathinones (SCs) represent one of the primary NPSs classes often seized in Europe, with use of the above-mentioned substances also for home consumption and/or alone [3]. SCs, commonly known as ‘bath salts’, are chemical derivatives of the natural khat plant (Catha edulis), which were first detected in Europe in 2008 [7].

SCs have gained high praise because they have been marketed as cheap substitutes for well-known psychostimulants amphetamine (AMPH), MDMA and cocaine [8,9], which, in addition to their purely stimulating effect, also induce pleasant effects, such as greater euphoria, empathy, sociability, increased libido and sexual performance [10,11]. However, SCs can be much more potent than the traditional psychostimulant drugs they imitate [12], and, therefore, the possibility of acute toxicity and death may be greater. To date, there are some substances, such as mephedrone (4-MCC), pentedrone and MDPV, that present both stimulating/empathogenic properties and stimulating/hallucinogenic effects [13,14]. Despite this, 2-((methylamino)methyl)-3,4-dihydronaphthalen-1(2H)-one or γ-mephtetramine (MTTA, Figure 1) turns out to be an atypical cathinone due to a further carbon between the amine and ketone group [15].

In fact, MTTA is a γ-aminoketone structurally similar to buphedrone or a mephedrone derivative, which is its positional isomer (Figure 1). Therefore, it should be assumed that MTTA exhibits a similar pharmacological and toxicological profile to one of these two aforementioned cathinones. Different from buphedrone, mephedrone is classified as MDMA–cocaine-like in terms of interaction with monoamine transporters [16,17,18]. However, information available in scientific literature about MTTA is scarce. Odoardi and her colleagues, through the first metabolism study performed in silico and followed by in vivo experiments, have recently shown that MTTA produces around 10 main metabolites in murine urine [19]. MTTA was first released and reported by the UK in 2013 [20], and in the same year it was also found in tools seized on Italian soil [21]. A peculiarity of MTTA was the fact that it emerged as quickly as it later disappeared from the clandestine drug market [22]. This coincides with a decrease in the number of new cathinones detected in those years and a simultaneous increase in other groups of stimulants and phenethylamines [23]. A previous study has already hypothesised that this trend could be due to limitations of substitutions that the skeleton of the cathinone can bear [24], hence the need to draw on other groups for synthesis of new active compounds. Therefore, these trends did not allow to better identify the properties of substances and provide important information about adverse effects in humans at the same time. Psychonaut websites such as bluelight (http://bluelight.org/, accessed on 5 December 2022) and land-der-träume (https://www.land-der-traeume.de/, accessed on 5 December 2022) report some anecdotes about MTTA use dating back to 2013 and describing it as completely ineffective [25] or as able to slightly increase talkativeness without having euphoric effects [26]. Conversely, other users report having consumed MTTA powder una tantum, indicating that it can provide a feeling of euphoria through progressive mental stimulation (between two and ten hours) that can extend until achieving a hallucinatory state. Therefore, they suggested that anyone trying MTTA should avoid driving or walking to a destination and opt for paid transportation options [27]. This evidence, reported on drug forums by users, implies that MTTA abuse could impair normal driving or working performance, as already ascertained for other different types of NPSs [28,29,30,31]. Thus, this could represent a severe issue for public health, indicating the urgent need to investigate the potential link between sensorimotor alterations and affected driving and working human skills, as previously suggested by preclinical studies [32,33].

Considering the poor preclinical in vivo evidence of MTTA, the aim of this study is to understand its pharmaco-toxicological potential. Given the occasional and sporadic use reported by testimonials from drug-forums, increasing dosages of MTTA (0.1–30 mg/kg) have been administered once a week for four consecutive weeks to mimic a typical pattern of human consumption. Specifically, effects induced by treatment with MTTA on sensorimotor (visual, acoustic and tactile) responses, physiological parameters (breath rate, core temperature and surface temperature), motor performance (time on rod, number of steps and mobility time) and grip strength have been investigated in CD-1 male mice through a selected battery of behavioural tests widely used for preclinical characterisation of new psychoactive molecules. To deepen the knowledge of its pharmaco-toxicological action, possible variations characterising the blood (Scil Vet abc + and Element RC devices) and urine (Aution Micro device) biochemical profiles and heart, spleen, kidney and liver damage have been also investigated.

## 2. Materials and Methods

### 2.1. Animals

A total of 32 male ICR (CD-1^®^) mice, 3–4 months old, weighing 25–30 gr (ENVIGO Harlan Italy, Italy; bred inside the Laboratory for Preclinical Research (LARP) of University of Ferrara, Italy) were group-housed (5 mice per cage; a floor area per animal of 80 cm^2^; a minimum enclosure height of 12 cm on a 12:12-h light-dark cycle (light on at 6:30 AM), temperature of 20–22 °C, humidity of 45–55% and were provided ad libitum access to food (Diet 4RF25 GLP; Mucedola, Settimo Milanese, Milan, Italy) and water. Experiments were performed during the light phase. The experimental protocol followed in the present study was in accordance with the new European Communities Council Directive of September 2010 (2010/63/EU), a revision of the Directive 86/609/EEC, and was approved by the Italian Ministry of Health (license n. 223/2021-PR and extension CBCC2.46.EXT.21) and the Ethics Committee of the University of Ferrara. According to the ARRIVE guidelines, all possible efforts were made to minimise the number of animals used, minimise the animals’ pain and discomfort and reduce the number of experimental subjects. Mice were divided into two groups: group one (G1) was used for behavioural test (vehicle and repeated MTTA treatment) and group two (G2) for biochemical analysis of biological samples and histological evaluation of heart, spleen and kidney (vehicle and repeated MTTA treatment). Experimental protocol was reported in Figure 2.

### 2.2. Drug Preparation and Dose Selection

The MTTA was purchased from LGC Standards (LGC Standards, Milan, Italy). The compound was initially dissolved in absolute ethanol (final concentration: 2%) and Tween 80 (2%) and brought to the final volume with saline (0.9% p/v NaCl). The solution made with ethanol, Tween 80 and saline was also used as the vehicle. Drug was administered by intraperitoneal route at a volume of 4 ul/g body mass. The range of drug doses of MTTA (0.1–30 mg/kg i.p.) was chosen using interspecies dose scaling [34]. Increasing doses of MTTA (0.1–30 mg/kg i.p.) were administered once a week for four consecutive weeks and a wash-out time of 7 days was considered as dosing interval. The present protocol aims at mimicking sporadic human consumption of MTTA (see Section 1).

### 2.3. Behavioural Tests

In the present study, effect induced by each dosage of MTTA on behavioural responses was investigated using a battery of tests widely used in studies of “safety-pharmacology” in rodents [35,36,37,38,39]. To reduce the number of animals used, mice were evaluated in functional observational tests carried out in a consecutive manner according to the following time scheme: observation of visual object responses (frontal and lateral view), overall tactile response (vibrissae, corneal and pinnae reflexes), acoustic response, visual placing response, breath rate, mobility time, core and surface temperature variations, time on rod, number of step and grip strength response. Behavioural tests were conducted in a thermostat-controlled (temperature: 20–22 °C, humidity: 45–55%) and light (150 lux) room with a background noise of 40 ± 4 dB. The apparatus for sensorimotor and physiological test consisted of an experimental chamber (350 × 350 × 350 mm) with black methacrylate walls and a transparent front door. During the week before the experiment, each mouse was placed in the box and handled (once a day) every other day, i.e., 3 times, to get used to both the environment and the experimenter. To avoid mice olfactory cues, cages were carefully cleaned with a dilute (5%) ethanol solution and rinsed with water. All experiments were performed between 8:30 AM to 2:00 PM and conducted in blind by trained observers working in pairs [36]. The behaviour of mice was videotaped by a camera (B/W USB Camera day&night with varifocal lens; Ugo Basile, Italy) placed at the top or on one side of the box and analysed offline by a different trained operator.

#### 2.3.1. Evaluation of the Visual Response

Visual response was verified by two behavioural tests which evaluated the ability of the animal to capture visual information when the animal is moving (the visual placing response) or stationary (the visual object response). Visual placing response test is performed using a tail suspension modified apparatus able to bring down the mouse towards the floor at a constant speed of 10 cm/s [36]. The downward movement of the mouse was videotaped by a camera (B/W USB Cameraday&night with varifocal lens; Ugo Basile, Italy) placed at the base of the tail suspension apparatus. Movies were analysed off-line by a trained operator who was unaware of the drug treatments performed, to evaluate the beginning of the reaction of the mouse approaching the floor. When the mouse started the reaction, an electronic ruler evaluated the perpendicular distance in millimetres between the eyes of the mice to the floor. Untreated control mice typically perceive the floor and prepare to contact at a distance of about 28 ± 4.3 mm. Tests were measured at 15, 35, 70, 125, 185, 245 and 305 min after the injection.

Visual object response test was performed to evaluate the ability of the mouse to see an object approaching from the front (frontal view) or the side (lateral view) that typically induces the animal to shift or turn the head or retreat from it. A white horizontal bar was moved frontally to the mouse head, while a small dentist’s mirror was moved laterally into the mouse’s field of view in a horizontal arc, until the stimulus was between the mouse’s eyes. The procedures were conducted bilaterally and repeated 3 times [36]. The score assigned to the movement was 1 or 0 if it was not present. The total value was calculated by adding the scores obtained in the frontal with those obtained in the lateral visual object response test (overall score: 9). Tests were measured at 10, 30, 60, 120, 180, 240 and 300 min after the injection.

#### 2.3.2. Evaluation of Tactile Response

Tactile responses were verified through vibrissae, corneal and pinnae reflexes induced by the touch of a thin hypodermic needle [36]. Data are expressed as the sum of the three above-mentioned parameters (overall score = 12). Vibrissae reflex was evaluated by touching vibrissae (right and left) once for side giving a value of 1 if there was a reflex (turning of the head to the side of touch or vibrissae movement) or 0 if not present (overall score: 2). Corneal reflex was assessed by gently touching the cornea of the mouse with a thin hypodermic needle and evaluating the response: the score assigned was 1 if the mouse moved only the head, 2 if it only closed the eyelid, 3 if it closed the lid and moved the head. The procedure was conducted bilaterally (overall score: 6). Pinna reflex was assessed by touching pavilions (left and right) with a thin hypodermic needle: first the interior pavilions and then the external. This test was repeated twice for side giving a score of 1 if a reflex was present and 0 if it was not present (overall score: 4). Sensorimotor tests were measured at 10, 30, 60, 120, 180, 240 and 300 min after the injection for the evaluation of the tactile response.

#### 2.3.3. Evaluation of Acoustic Response

Acoustic response measures the reflex of the mouse in response to an acoustic stimulus produced behind the animal [36]. Four acoustic stimuli of different intensity and frequency were tested: a snap of the fingers (four snaps repeated in 1.5 s), a sharp click (produced by a metal instrument; four clicks repeated in 1.5 s), an acute sound (produced by an audiometer; frequency: 5.0–5.1 kHz) and a severe sound (produced by an audiometer; frequency: 125–150 Hz). Each test was repeated 3 times. The score assigned was 1 if there was a response or 0 if it was not present, for a total score of 3 for each sound (overall score: 12). The background noise (about 40 ± 4 dB) and the sound from the instruments were measured with a digital sound level meter. Sensorimotor tests were measured at 10, 30, 60, 120, 180, 240 and 300 min after the injection.

#### 2.3.4. Evaluation of Breath Rate

The experimental protocol for the detection of respiratory parameter in this study provides for monitoring of the animal awake, freely moving, with a non-invasive and minimal handling. The animal is leaving free in a cage and the respiration patterns of the mice were videotaped by a camera (B/W USB Camera day & night with varifocal lens; Ugo Basile, Italy) placed above observation’s cage. A trained operator who does not know the drug treatments performed analyses movies off-line. The analysis frame by frame allows to better evaluate the number of breath rates of the mouse evaluated through the count of about 257 ± 11 breath rates per minutes (brpm). Breath rate was measured at 15, 40, 70, 150, 130, 190, 250 and 310 min after the injection [39].

#### 2.3.5. Evaluation of Surface and Core Temperature

To better assess the effects of the ligands on thermoregulation, we measured changes in the surface and core (rectal) temperature. Surface temperature was measured with an infrared thermometer. The core temperature was evaluated by a probe (1 mm diameter) that was gently inserted, after lubrication with liquid Vaseline, into the rectum of the rat (to about 2 cm) and left in position until the stabilization of the temperature (about 10 s) [35]. The probe was connected to a Cole Parmer digital thermometer, model 8402. Core temperature was measured at 30, 50, 85, 140, 200, 260 and 320 min after the injection.

#### 2.3.6. Motor Activity Assessment

Stimulated and spontaneous motor activity alterations were measured performing the Accelerod test, Drag test and the Mobility time test [35,36,40].

The Accelerod test evaluate motor coordination, locomotor ability (akinesia/bradykinesia), balance ability, muscular tone and motivation to run. To this end, the animals were placed on an accelerod apparatus (Ugo Basile SRL, Gemonio [VA], Italy) that automatically increases rotation speed in a constant manner (0–60 rotations/min in 5 min). The time spent on the cylinder was measured [35,40,41]. The accelerod test was performed at 40, 65, 95, 150, 210, 270 and 330 min after the injection.

The Drag test evaluate the ability of the animal to balance the body posture with the front legs in response to an external dynamic stimulus [40,41]. The mouse was lifted by the tail, leaving the front paws on the table and dragged backward at a constant speed (20 cm/s) for a fixed distance (100 cm). The number of steps performed by each paw was recorded by two different observers. For each animal from five to seven measurements were collected [35]. The drag test was performed at 45, 70, 105, 160, 220, 280 and 340 min after the injection.

The Mobility time test evaluate spontaneous motor activity of mice [42]. The mouse is free to move on a square plastic cage (60 × 60 cm). The observer measures the total time spent moving by the animal (when the mouse walks or moves the front legs) in five minutes. Test was performed at 15, 35, 70, 125, 185, 245 and 305 min after the injection.

#### 2.3.7. Evaluation of Skeletal Muscle Strength (Grip Strength)

Grip strength test was used to evaluate the skeletal muscle strength of mice [38,39]. The grip strength apparatus (ZP-50 N, IMADA) is made of a wire grid (5 × 5 cm) connected to an isometric force transducer (dynamometer). The test was performed by holding the mice by the tails and allowed them to grasp the grid with their forepaws. Then, mice were gently pulled backward until the release of the grid. The average force exerted by each mouse before losing the grip was recorded. The mean average force was then determined calculating the mean of three measurements for each animal. The skeletal muscle strength is expressed as gram force (gf) and processed using IMADA ZP-Recorder software. Grip strength was measured at 0, 45, 70, 105, 160, 220, 280 and 340 min after the injection.

### 2.4. Biochemical Studies

#### 2.4.1. Collection of Samples

The samples were collected at the end of the present study, which provides for the repeated administration of MTTA (0.1–30 mg/kg, i.p.). Specifically, blood and urine samples were collected and analysed after the last injection (Figure 2; 30 mg/kg, i.p.).

Urine specimens were collected from mice individually placed inside metabolic cages (Ugo Basile SRL, Gemonio [VA], Italy) with free access to water and food [43,44,45]. To evaluate a potential urine chemistry variation, Aution Micro instrument, that allows to measure chemical-physical parameters (SCIL Animal Care Company, Treviglio, Italy), has been used. Following the urine samples collection, blood samples were collected by submandibular blood collection technique. At the end of this procedure, the animals have been sacrificed. After each blood withdrawal, the total sample volume has been split transferring it into 1-mL vial containing ethylendiaminetetracetic acid (EDTA, as preservative and anticoagulant) and analysed through the scil Vet abc Plus+ and into HEPARIN tube and analysed through Element Rc.

#### 2.4.2. The Scil Vet abc Plus+

The scil Vet abc Plus™ is a simple (SCIL Animal Care Company, Treviglio, Italy) haematology analyser with a micropipette aspirator that needs a sample of 10 μL (containing EDTA as anticoagulant preservative) to produce a differential and complete blood cell count: erythrocytes, thrombocytes, leukocytes, haemoglobin and all erythrocyte-indices. 4-Part Differential: Granulocyte, Monocyte, Lymphocyte, Eosinophil. blood count [46]. Sample analysis provides for the quantifying of the following parameters: White Blood Cells (WBC), Lymphocytes (LYM), Monocytes (MON), Granulocytes (GRA), Eosinophils (EOS), Red Blood Cells (RBC), Haemoglobin (HGB), Haematocrit test (HCT), Mean Corpuscular Volume (MCV), Mean Corpuscular Haemoglobin (MCH), Mean Corpuscular Haemoglobin Concentration (MCHC), Red cell Distribution Width (RDW), Platelets (PLT), Mean Platelet Volume (MPV). The measurement is started by pushing one button and results are automatically transferred to the practice software program. The operator-friendly reagent pack concept renders daily maintenance procedures unnecessary. The proven impedance technology delivers reliable results in both healthy and sick animals.

#### 2.4.3. Element RC

The Element RC is an absorption spectroscopy clinical chemistry analyser that ensures high accuracy. The clinical chemistry analyser is feature by a rotor-based chemistry (Comprehensive rotor) or electrolyte (Electrolyte rotor) solution for a superior diagnostic capability. Within 12 min you receive results of 15 parameters. As regard to Comprehensive parameters these include Albumin (ALB), Total Protein (TP), Globulin (GLOB), Albumin/Globulin Ratio (A/G), Total Bilirubin (TB), Gamma Glutamyltransferase (GGT), Alanine Aminotransferase Level (ALT), Alcaline Phosphatase Level (ALP), Amylase (AMY), Creatinine (Crea), Triglycerides (TG), Glucose (GLU), Calcium (Ca), Phosphate (PHOS), Blood Urean Nitrogen/Creatinine (BUN/Crea) and Blood Urean Nitrogen (BUN). On the other hand, Electrolyte parameters include total CO2 (tCO2), Calcium (Ca), Phosphates (PHOS), Magnesium (Mg), Potassium (K+), Sodium (Na+) and Chlorine (Cl-). The Element RC requires only 100 µL lithium heparin sample of plasma for a diagnostic profile. The Element RC supports a bidirectional data transmission to the practice management software. Pre-installed analyser specific reference ranges and results can be transmitted directly into your practice management software.

#### 2.4.4. Aution Micro

The Aution Micro is an analyser for the chemical-physical analysis of urine and allows to evaluate up to 10 chemical-physical parameters: Glucose (GLU), Protein (PRO), Bilirubin (BIL), Urobilinogen (URO), Ph, Specific Gravity (S.G.), Blood (BLD), Ketones (KET), Nitrates (NIT) and Leucocytes (LEU). The Aution Micro analyser provides a semi-quantitative evaluation of the proteinuria/creatinuria PU/CU ratio without any sample preparation step (dilution/centrifugation). In fact, urine analysis is performed by using reactive strips made for specific diagnostic needs (Aution Sticks 9UB, Aution Sticks 10EA, Aution Sticks 10PA and Aution Sticks Microalbuminuria/creatinuria).

### 2.5. Histological Studies

#### 2.5.1. Collection of Tissue Samples

The samples of heart, spleen, liver and kidney of the two groups of animals (vehicle-treated and MTTA-treated) were collected. The organs of MTTA- and vehicle-treated animals were immediately removed, weighed and placed in 10% buffered formalin for 48 h, then embedded in paraffin. Paraffin-embedded tissue specimens were sectioned at 5 μm, and then haematoxylin and eosin stain were performed.

#### 2.5.2. Histological Procedure

For histological investigations, the sections were stained with haematoxylin and eosin and then observed under optical microscope (Nikon Eclipse E90i; Nikon, Roma, Italy).

### 2.6. Statistical Analysis

In sensorimotor response experiments, data are expressed in arbitrary units (visual objects response, acoustic response, vibrissae, corneal and pinnae reflex) and percentage of baseline (visual placing response, breath rate, mobility time, drag, accelerod tests and grip strength). Core temperature values were expressed as the difference between control temperature (before injection) and temperature following drug administration (Δ °C) for each substance and presented in maximal possible effect:EMax% = [(test T° − ontrol T°)**/**(cut off T° − control T°)] × 100

All data are shown as mean ± SEM of 8 independent experimental replications. Statistical analysis of the effects of each compound at different concentrations over time was performed by two-way ANOVA followed by Bonferroni post hoc test for multiple comparisons. In the biochemical study, the statistical analysis of the MTTA (30 mg/kg) effects was performed using Student’s *t*-test used to compare vehicle and treated. For all tests, a value of *p* < 0.05 was considered statistically significant. All statistical analyses were performed using GraphPad Prism 8 software for Windows (La Jolla, CA, USA).

## 3. Results

### 3.1. Behavioural Studies

#### 3.1.1. Evaluation of the Sensorial Response

Visual placing, visual object, acoustic and overall tactile response did not change in vehicle-treated mice (Figure 3), and the effect was similar to that observed in naïve untreated animals (data not shown).

Visual Placing Test. Systemic administration of MTTA (0.1–10 and 30 mg/kg; i.p.) dose-dependently reduced the visual placing response in mice (Figure 3, panel A). The highest dose (30 mg/kg) induced deep and prolonged inhibition of visual placing, while intermediate doses (1 and 10 mg/kg) induced transient inhibition up to ~50% and 60%, respectively (significant effect of treatment (F4,280 = 240.8, *p* < 0.0001), time (F7,280 = 47.61, *p* < 0.0001) and time × treatment interaction (F28,280 = 8.759, *p* < 0.0001).

Visual Object Test. MTTA at the highest dose tested (30 mg/kg, i.p.), reduced Visual object response during the first hour of experiment ~18% compared to basal values, while the 10 mg/kg dose induced a mild reduction of ~12% only during the first 5 min (Figure 3, panel B; significant effect of treatment (F4,280 = 10.55, *p* < 0.0001), time (F7,280 = 5.325, *p* < 0.0001); significant effect of treatment (F4,280 = 10.55, *p* < 0.0001).

Overall Tactile Reflex Test. All MTTA doses (0.1, 1, 10 and 30 mg/kg i.p. Figure 3, panel C) did not induce a significant effect on overall tactile reflex test.

Start Reflex Test. Only at the highest dose (30 mg/kg, i.p.), MTTA slightly decreased acoustic response ~11% at 60 min (Figure 3, panel D, significant effect of treatment (F4,280 = 8.453, *p* < 0.0001), time (F7,280 = 1.824, *p* = 0.0826) and time x treatment interaction (F28,280 = 1.196, *p* = 0.2333).

#### 3.1.2. Evaluation of the Physiological Parameters

Breath rate, core and surface temperature did not change in vehicle-treated mice over the 5 h observation (Figure 4), and the effect was similar to that observed in naïve untreated animals (data not shown).

Core Temperature. Core temperature immediately dropped ~4 °C compared to basal after MTTA (30 mg/kg, i.p.) administration. The thermic response gradually returns to basal values from 50 min (Figure 4, panel B; significant effect of treatment (F4,245 = 5.874, *p* = 0.0002), time (F6,245 = 1.660, *p* = 0.1315) and time x treatment interaction (F24,245 = 3.250, *p* < 0.0001). Other doses did not provoke effects on core temperature.

Surface Temperature. 30 mg/kg (i.p.) of MTTA slightly reduced surface temperature ~2 °C only at 5 min of experiment (Figure 4, panel C; effect of treatment (F4,245 = 1.819, *p* = 0.1258), time (F6,245 = 0.5575, *p* = 0.7639) and time x treatment interaction (F24,245 = 1.020, *p* = 0.4404). Other doses did not provoke effects on core temperature.

#### 3.1.3. Evaluation of the Motor Response

Time on rod, number of steps and grip strength remained unchanged in vehicle-treated mice over the 5 h observation (Figure 5, panels A, B and D), and effect was similar to that observed in naïve untreated animals (data not shown).

Mobility time decreased in vehicle-treated mice over the 5 h of experiment (Figure 5, panel C) as with untreated animals (data not shown).

Accelerod Test. Systemic administration of MTTA (0.1–10 and 30 mg/kg, i.p.) dose-dependently and long-lastingly increased the stimulated motor activity in mice, which was monitored by accelerod test. In particular, after 1, 10 and 30 mg/kg administration, the time on rod increased to ~20%, ~30% and ~40% with respect to basal (Figure 5, panel A, significant effect of treatment (F4,280 = 34.93, *p* < 0.0001), time (F7,280 = 5.301, *p* < 0.0001) and time × treatment interaction (F28,280 = 1.833, *p* = 0.0078).

Drag Test. Systemic administration of MTTA induced significant number of steps changes at 0.1, 10 and 30 mg/kg (Figure 5, panel B, significant effect of treatment (F4,280 = 15.68, *p* < 0.0001), time (F7,280 = 20.44, *p* < 0.0001) and time x treatment interaction (F28,280 = 5.036, *p* < 0.0001). MTTA at 0.1 mg/kg slightly and constantly increased the number of steps up to the end of the experiment. Administration of 10 mg/kg MTTA slightly increased the number of steps only during the last 2 h of observation period. Finally, the highest dose of MTTA (30 mg/kg, i.p.) induced a biphasic effect, decreasing the number of steps during the first 105 min and increased it during last hours.

Mobility Time Test. The highest dose (30 mg/kg, i.p.) of MTTA induced a biphasic effect on spontaneous motor activity, which initially is rapidly decreased ~40% with respect to basal in the first 15 min and then increased ~40% during the last 2 h (Figure 5, panel C, significant effect of treatment (F4,278 = 1.978, *p* = 0.0981), time (F7,278 = 5.510, *p* < 0.0001). Other doses did not induce a different effect from basal.

Grip strength. The dose–response curve did not show any significant effect of skeletal muscle strength after systemic administration of MTTA at all doses (0.1–10 and 30 mg/kg, i.p.; Figure 5, panel C).

### 3.2. Biochemical Studies

#### 3.2.1. Complete Blood Count

Vehicle-treated mice did not show blood count parameters variation compared to naïve untreated control mice (data not shown). Systemic administration of MTTA induced significant variation in blood count parameters. In fact, Table 1 showed significant decrease following *t*-test analysis: WBC (t = 17.16, Df = 14, *p* < 0.0001), LYM# (t = 9.649, Df = 14, *p* < 0.0001), LYM% (t = 16.78, Df = 14, *p* < 0.0001), RBC (t = 15.37, Df = 14, *p* < 0.0001), HGB (t = 25.73, Df = 14, *p* < 0.0001), HCT (t = 15.90, Df = 14, *p* < 0.0001), MCV (t = 51.90, Df = 14, *p* < 0.0001), MCH (t = 37.91, Df = 14, *p* < 0.0001), MCHC (t = 26.66, Df = 14, *p* < 0.0001), RDW (t = 10.37, Df = 14, *p* < 0.0001), MPV (t = 22.42, Df = 14, *p* < 0.0001). On the contrary, MON% (t = 2.164, Df = 14, *p* = 0.0482) and GRA% (t = 4.508, Df = 14, *p* = 0.005) increased.

#### 3.2.2. Element RC

MTTA induced changes in electrolyte (Table 2) and comprehensive (Table 3) parameters in mice, whereas parameters of vehicle-treated mice did not change compared to naïve mice (data not shown).

#### 3.2.3. Urine Analysis

Urine parameters did not change in vehicle-treated mice compared to untreated mice (data not shown). Systemic administration of MTTA (Table 4) induced an increase in following parameters: PRO ( t = 5.036, Df = 14, *p* = 0.0002), URO (t = 16.10, Df = 14, *p* < 0.0001), pH (t = 2.579, Df = 14, *p* = 0.0218), BLD (t = 8.944, Df = 14, *p* < 0.0001), KET (t = 5.837, Df = 14, *p* < 0.0001) and LEU (t = 10.19, Df = 14, *p* < 0.0001).

### 3.3. Histological Results

#### 3.3.1. Kidney

Glomeruli showed fragmentation, and few atrophied elements were observed. Perihilar segmental sclerosis and collapsing lesion relating to hypertrophied elements were evident and diffuse in the same glomeruli. There were small adhesions to the Bowman’s capsule. Patchy mild interstitial leucocytes infiltration involving the cortex was described (Figure 6, panels A and B). Pathological findings were absent in the vehicle-treated mice (not shown). 

#### 3.3.2. Heart

In the heart samples, the histologic changes are an intense hyper-eosinophilia of the hypercontracted myocardial cells with rhexis of the myofibrillar apparatus into cross-fibre, anomalous and irregular or pathological bands (contraction bands necrosis). Normal cells around hypercontracted ones assume a wavy appearance. The spaces between bands are filled by mitochondria. No evidence of platelet aggregation or other vessel changes or of interstitial or sarcolemmal alterations exists (Figure 6, panels C and D). No pathological findings were present in the vehicle-treated mice (not shown).

#### 3.3.3. Spleen

The analysis of spleen tissue does not show pathological findings either in treated or vehicle-treated mice (not shown).

#### 3.3.4. Liver

Liver samples of treated mice show moderate initial cytoplasmic hyper-eosinophilia, foci of necrosis, sinusoidal dilatation with presence of acidophilic body and ballooning degeneration of hepatocytes (Figure 6, panels E and F). In periportal areas in preserved hepatocytes were observed microvescicular fatty changes. In the vehicle-treated mice were not present histological alterations (not shown).

## 4. Discussion

The aim of this study is to evaluate the profile of amphetamine derivate MTTA, a new psychoactive substance (NPS) belonging to the synthetic cathinones (SCs) group [3]. Structurally, SCs are β-keto amphetamines derivates and elicit their action on monoamines transporters [47,48,49,50]. The tested substance, MTTA, is a cathinones-like compound, but it presents a γ-amino ketonic structure [15,19]. Its structure, different from that of β-keto-derivates, could explain why MTTA did not present the typical profile when compared to other already studied synthetic stimulants [51,52,53]. For the first time, our study shows that repeated MTTA administration induces behavioural, biochemical and histological changes (in heart, liver and kidney) in mice.

### 4.1. Behavioural Changes

Visual placing test demonstrated a significant decrement after 1, 10 and 30 mg/kg (Figure 3, panel A) of MTTA administration, while visual object test showed a transient decrease after 30 mg/kg administration (Figure 3, panel B). Acoustic reflexes were slightly reduced at 30 mg/kg (Figure 3, panel D). As regards physiological responses, MTTA transiently reduced breath rate at the highest dose (30 mg/kg, Figure 4, panel A). MTTA at 30 mg/kg induced core and surface temperature decrease during the first 50 min (Figure 4, panels B and C). Motor changes were reported after MTTA administration (Figure 5) as follows: Time on rod was dose-dependently increased (panel A), while biphasic effect was shown at the highest dose (30 mg/kg) in drag test (panel B) and mobility time test (panel C). Finally, performing grip strength test, any change in the skeletal muscle strength of mice was not observed after MTTA administration (Figure 5, panel D).

The deepest MTTA effect, observed in visual placing test, was likely caused by norepinephrine transporter (NET) inhibition. Such inhibition could lead to visual placing responses decrease, as previously demonstrated by De-Giorgio and colleagues’ behavioural study on amphetamine-derivate methiopropamine (MPA) [52]. In particular, the norepinephrine (NE) release could change optokinetic reflex (OKR) via β-receptor and vestibulo ocular reflex (VOR) via α2-receptors [54] stimulation. The vestibular system is the main circuit involved in visual placing test, ruling visual functions, postural equilibrium and complex voluntary motions [55]. Indeed, together with visual cues, motor and postural circuits could be involved in response to proprioceptive information [56]. The same hypothesis was generated regarding the above-mentioned MPA study [52]. Long lasting effects on visual placing could also be due to MTTA metabolites’ action on monoaminergic transporters [19].

Despite MTTA’s likely noradrenergic profile, even the hypothesis of serotoninergic system involvement should be considered. MTTA could also increase serotoninergic tone, inhibiting serotonin transporter (SERT). Indeed, activation of 5HT2 receptor in cortico-visual circuits could explain the visual object responses, probably due to a “disperceptive state” of animals [42]. Likewise, the acoustic reflex decrement at the highest dose tested (30 mg/kg) could be due to serotonin action on the dorsal nucleus accumbens, in which 5HT2 binding could lead to auditory suppression [57]. This evidence is typical of entactogenic substances, such as MDMA [18], that, at the highest dose tested (20 mg/kg), induced the same effect on mice [42]. Moreover, even mephedrone has been reported to induce visual and auditory hallucination in humans [58].

Further confirmation of the involvement of the serotoninergic system is given by seizures noted in tested mice during the first hour of observation (12.5% of mice at 10 mg/kg and 37.5% of mice at 30 mg/kg; data not shown). Breath rate decrease co-appears in mice in the same period of convulsive events. Seizures, deriving from amygdala stimulation [59], could provoke a reduction in spontaneous breathing [60]. During a convulsion, a specific amygdala region could be stimulated [61]. This presents different effector sites downstream, such as the breathing central pattern generator (bCPG), that resides in the pre-Bötzinger complex (pre-BötC), which mediates respiratory rhythm [62,63]. MTTA could also regulate body temperature, modulating monoamines tone in the preoptic-anterior hypothalamic area, which is involved in thermoregulation [64,65]. Hypothermia could be again induced through binding of NE to α1-receptor [66]. Moreover, within the same brain region, even DA release, deriving from dopaminergic transporter (DAT) inhibition, could provoke the reduction in temperature, as previous studies on rats have shown [67]. The MTTA-induced effect on temperature was similar but less long-lasting when compared to that of mephedrone in rats in the same experimental condition [68]. In addition, MPA in mice or other amphetamine derivates such as methamphetamine (core temperature) [52,69] and MDPV in rats (surface temperature) [70] induced hypothermic response. DAT inhibition could also explain the dose-dependent locomotor activity increase (observed in accelerod test, Figure 5, panel A), which is consistent with a previous preclinical in vivo study that showed these effects in rodents after administration of different cathinones, including mephedrone and buphedrone [52,71,72,73,74]. This evidence suggests that DAT inhibition could lead to a dopaminergic tone increment within nucleus accumbens [75,76,77,78]. The same evidence was reported after buphedrone administration, resulting in increased D1 receptor expression in dorsal striatum and nucleus accumbens [72]. Nevertheless, two other motor tests (drag and mobility time test, Figure 5, panel B and C) showed initial inhibition of motor activity at the highest dose tested (30 mg/kg). Once again, an appearance of seizures may explain the effect. Indeed, malaise of mice may explain the initial decrease in motor activity since motor facilitation resumed after the disappearance of convulsions. Another explanation of biphasic effect could be based on MTTA metabolism [19], which could produce a metabolite that mostly acts on dopamine, promoting motor facilitation. Moreover, due to the absence of effect induced on skeletal muscle strength, it could be excluded that the initial motor decrease was related to an impairment of skeletal muscle activity.

It is worth noting that mephedrone, which is similar to mephtetramine in its chemical structure, causes efflux of dopamine and serotonin via acting on monoamines’ transporters, with similar inhibitory efficacy and potency [79]. Thus, based on the present results, a similar pharmacological profile should be suggested for MTTA. However, further studies may be required to deepen insight regarding the mechanisms underlying its atypical effects.

### 4.2. Biochemical and Histological Changes

Alterations in the biochemical profile of blood and urine show for the first time the potential toxicity of repeated administration of MTTA in mice. In fact, many authors have proven that investigating mouse models can be useful to investigate blood disorders in humans [80]. Concerning blood parameters, some of these could have changed due to haemorrhage [81,82] and stress status [82] provoked by submandibular withdrawal or heparinised sampling [83]. Specifically, the latter evidence was previously related to electrolyte levels variation in preclinical studies, such as Ca [82,84] or Mg [82,85,86]. Despite this, the same withdrawal and sampling technique was used for both basal sample (vehicle-treated animals) and treated sample (MTTA-treated animals), and the first one reported values similar to those of naïve untreated mice (data not shown). Therefore, we assume that blood parameters variation could be closely related to drug administration. In particular, through the complete blood count (Table 1), the mice treated with MTTA showed variations in parameters in respect to vehicle-treated animals. In fact, there was an anaemia and leukopenia with monocytosis and granulocytosis. In particular, we demonstrated a decrease in WBC, LYM, RBC, HGB, HCT, MCV, MCH, MCHC, RDW and MPV, while there was an increase in MON and GRA. The simultaneous decrease in lymphocytes together with a relative increase in the monocytes and granulocytes is compatible with an immediate inflammatory response following administration of last MTTA dose (30 mg/kg). On the other hand, electrolytes parameters changes showed a decrease in Ca, Phos, Mg and Na, whereas tCO2 increased (Table 2). The increase in tCO2 value could indicate hypoventilation, confirmed by the decrease in breath rate (Figure 4, panel A) following MTTA administration. Other blood parameters’ analysis indicates variation in ALB, TP, A/G, TB, GGT, ALT, ALP, AMY, Crea, TG, GLU and Ca2+(Table 3). In line with histological analysis, the variation in ALT, ALP, GGT, TB and ALB could lead to a liver disease [87,88]. The lower ALB than basal value could also explain the TP and A/G decrease. Pancreatic disorder could be suggested by the increase in TC and GLU [89,90]. The reduction in AMY could be related to these changes, being an index of metabolic syndrome and diabetes [91].

Finally, urine samples showed an increase in PRO, URO, PH, BLD, KET and LEU (Table 4). Bilirubin and urobilinogen in urine could confirm the possibility of liver damage [92]. In particular, the intestinal metabolism of bilirubin converts this to urobilinogen, which is excreted with urine [93]. In addition, the pH change could indicate a metabolic alteration, leading to an acid–base imbalance that could confirm the hypoventilation described. Therefore, the urinary profile indicates an important renal failure, with the impossibility of acid–base recovery established by two possible mechanisms: decreased breath rate with increased blood carbon dioxide and/or direct metabolic reaction to metabolization of the drug. Histological analyses show evident renal impairment at the glomerular level and Bowman’s capsule and mild interstitial leucocytes infiltration involving the renal cortex. This kidney damage could be the cause of high protein urine levels [94]. Moreover, the presence of ketones in urine is an index of high glucose blood levels [95], confirming possible metabolic damage already reported. Possible damage in these organs could also explain the very high level of leucocytes, which is an index of inflammation [96]. Cardiac alterations characterised by intense hyper-eosinophilia of the hypercontracted myocardial cells with rhexis of the myofibrillar apparatus into cross-fibre, anomalous and irregular or pathological bands (contraction bands necrosis) indicate a state of stress and suffering at the level of this organ and further suggest the dangers of repeated intake of MTTA.

## 5. Conclusions

The present study pointed out for the first time that atypical cathinone MTTA induces sensorial (inhibition of visual and acoustic reflexes) and transient physiological parameter (decrease in breath rate and body temperature) changes in mice. Regarding motor activity, both a dose-dependent increase (accelerod test) and biphasic effect (drag and mobility time test) have been shown. The structural profile different from that of the most common cathinones [12,49,51,70,97,98] could suggest that MTTA has been synthetised with the aim to induce different effects from previous drugs, also in line with the mild stimulant profile highlighted by the present results. Thus, MTTA users showed different effects [26,99,100] compared to those expected, and this may explain why the substance was not used for a long time but was available in the market only from 2013 to 2015 [22]. Despite this mild cathinone-like effect, MTTA induced important changes in both blood and urine profile, possibly consistent with trigger of inflammatory responses and/or metabolic alterations, emphasizing its potential toxicity, strengthened by the observed histological changes in heart and kidney samples. Therefore, MTTA, even if perceived as not very active and relatively safe by consumers, hides a potential toxicity that must be further considered, especially when MTTA is taken together with other and more powerful NPSs.

## Figures and Tables

**Figure 1 brainsci-13-00161-f001:**
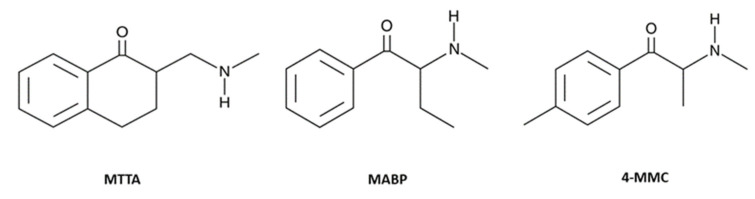
Chemical structures of γ-mephtetramine (MTTA), buphedrone (MABP) and mephedrone (4-MMC) from the Cayman Chemical website (https://www.caymanchem.com, accessed on 25 November 2022, at 1.30 p.m.).

**Figure 2 brainsci-13-00161-f002:**
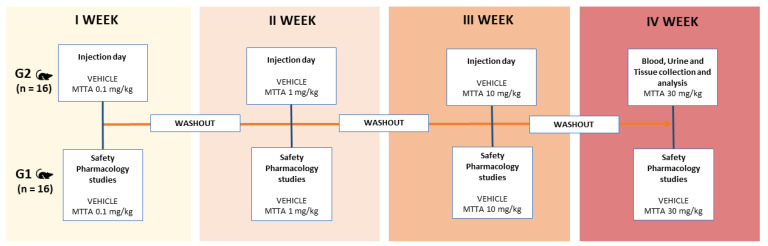
Schematic representation of protocol used for the experiments. Mice were divided into two groups and treated for 4 consecutive weeks (once a week) with vehicle or increasing doses of MTTA (0.1–30 mg/kg; i.p.). Group one (G1) was employed for safety pharmacology test, performed following each injection. Group two (G2) was employed for biochemical and histological analysis.

**Figure 3 brainsci-13-00161-f003:**
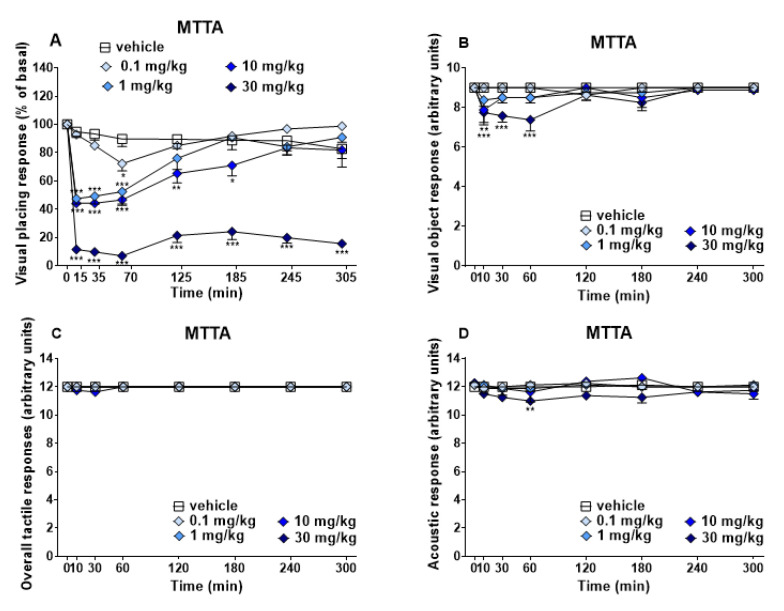
Effects of MTTA (0.1–30 mg/kg i.p.) on the visual placing response (panel (**A**)), visual object response (panel (**B**)), overall tactile response (panel (**C**)) and start reflex response (panel (**D**)) of mice. For visual placing test, data are expressed as percentage of baseline and represent the mean ± SEM of 8 mice for each treatment. For visual object response, overall tactile and start reflex tests, data are expressed as arbitrary units and represent the mean ± SEM of 8 mice for each treatment. Statistical analysis was performed by two-way ANOVA followed by Bonferroni’s test for multiple comparison. * *p* < 0.05; ** *p* < 0.01; *** *p* < 0.001 versus vehicle.

**Figure 4 brainsci-13-00161-f004:**
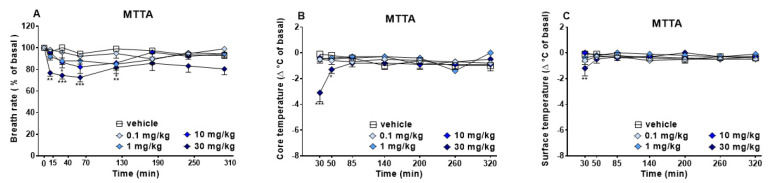
Effects of MTTA (0.1−30 mg/kg i.p.) on the breath rate (panel (**A**)), core temperature (panel (**B**)) and surface temperature (panel (**C**)) of mice. Breath rate values are expressed as percentage of baseline and represent the mean ± SEM of 8 mice for each treatment. Core and surface temperature are expressed as Δ °C of baseline and represent the mean ± SEM of 8 mice for each treatment. Statistical analysis was performed by two-way ANOVA followed by Bonferroni’s test for multiple comparison. * *p* < 0.05; ** *p* < 0.01; *** *p* < 0.001 versus vehicle. Breath rate. Systemic administration of the highest dose (30 mg/kg, i.p.) reduced breath rate ~ 30% with respect to basal during the first 3 h of experiment, while intermediate doses (1 and 10 mg/kg, i.p.) slightly reduced breath rate only at 130 and 40 min, respectively (Figure 4, panel (**A**); significant effect of treatment (F4,280 = 18.93, *p* < 0.0001), time (F7,280 = 5.714, *p* < 0.0001) and time × treatment interaction (F28,280 = 1.273, *p* = 0.1657).

**Figure 5 brainsci-13-00161-f005:**
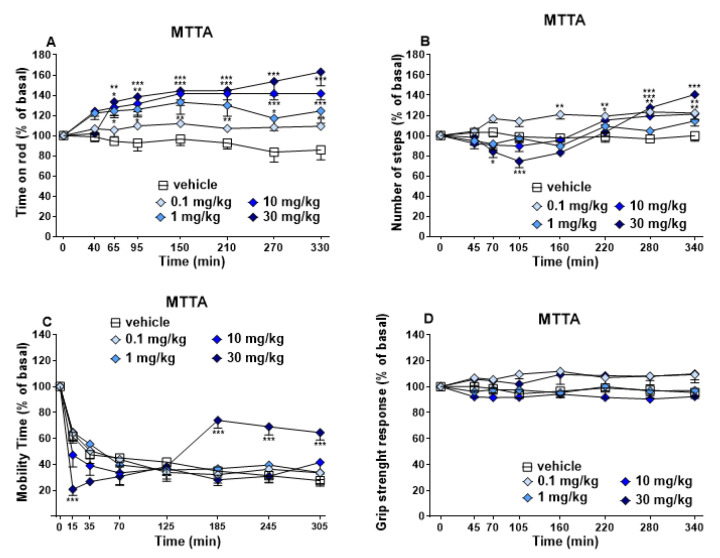
Effects of MTTA (0.1–30 mg/kg i.p.) on the time on rod (panel (**A**)), number of steps (panel (**B**)), mobility time (panel (**C**)) and grip strength (panel (**D**)) of mice. Data are expressed as percentage of baseline and represent the mean ± SEM of 8 mice for each treatment. Statistical analysis was performed by two-way ANOVA followed by Bonferroni’s test for multiple comparison. * *p* < 0.05; ** *p* < 0.01; *** *p* < 0.001 versus vehicle.

**Figure 6 brainsci-13-00161-f006:**
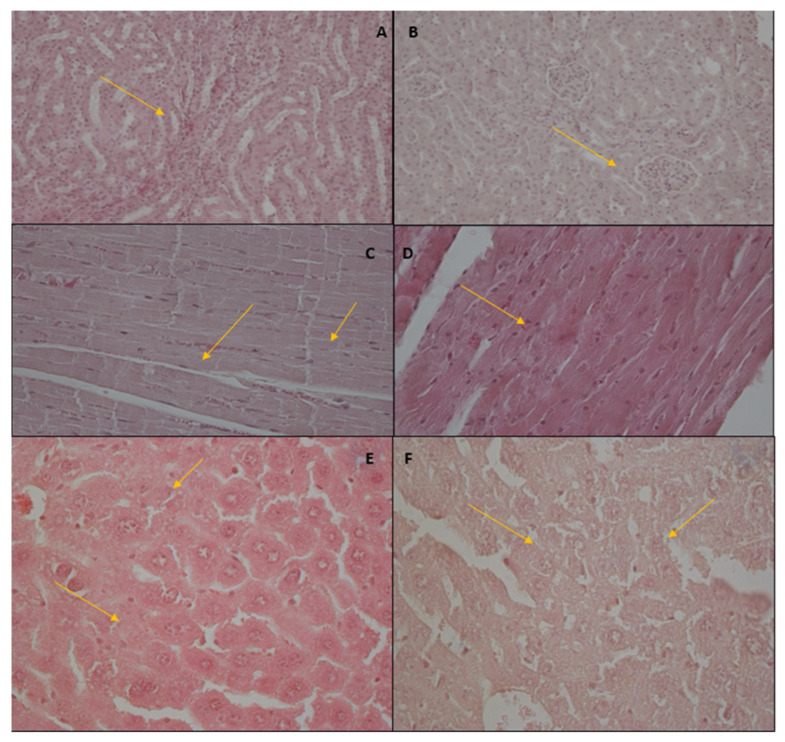
Histological aspects of kidney (**A**,**B**), heart (**C**,**D**) samples and liver (**E**,**F**) in MTTA-treated mice. Kidney: interstitial leucocytes infiltration (see arrows) and segmental sclerosis and collapsing lesion in glomeruli (see arrows). Heart: areas of contraction bands necrosis (see arrows). Liver: hepatocytes hyper-eosinophilia, foci of necrosis, sinusoidal dilatation (see arrows).

**Table 1 brainsci-13-00161-t001:** Effect of MTTA on blood count (WBC, LYM#, MON#, GRA#, EOS#, LYM%, MON%, GRA%, EOS%, RBC, HGB, HCT, MCV, MCH, MCHC, RDW, PLT, MPV) in mice. The data shown here (see Materials and Methods) refer to the mean ± SEM values relating to eight animals for vehicle and eight for MTTA. Student’s *t*-test was used to determine statistical significance (*p* < 0.05) between the two groups. * *p* < 0.05; ** *p* < 0.01; *** *p* < 0.001 versus vehicle.

Blood Count Parameters	Vehicle	MTTA	Variations
WBC (10^3^/mm^3^)	5.02 ± 0.234	3.30 ± 0.16 ***	⇓⇓
LYM# (10^3^/mm^3^)	2.68 ± 0.279	1.63 ± 0.13 ***	⇓⇓
MON# (10^3^/mm^3^)	0.26 ± 0.055	0.29 ± 0.02	-
GRA# (10^3^/mm^3^)	1.38 ± 0.259	1.39 ± 0.11	-
EOS# (10^3^/mm^3^)	0.36 ± 0.10	0.31 ± 0.10	-
LYM% (%)	58.12 ± 2.52	38.55 ± 2.13 ***	⇓⇓
MON% (%)	6.12 ± 0.34	6.75 ± 0.75 *	⇑
GRA% (%)	24.64 ± 2.31	29.52 ± 2.01 **	⇑
EOS% (%)	6.51 ± 1.35	5.54 ± 1.69	-
RBC (10^6^/mm^3^)	8.03 ± 0.17	6.25 ± 0.28 ***	⇓
HGB (g/dL)	15.92 ± 0.27	10.75 ± 0.50 ***	⇓⇓
HCT (%)	45.40 ± 1.23	34.19 ± 1.57 ***	⇓
MCV (μm3)	50.11 ± 0.38	42.90 ± 0.10 ***	⇓
MCH (pg)	17.69 ± 0.29	13.62 ± 0.09 ***	⇓
MCHC (g/dL)	31.50 ± 0.68	24.87 ± 0.18 ***	⇓
RDW (%)	12.64 ± 0.23	11.34 ± 0.27 ***	⇓
PLT (10^3^/mm^3^)	583.78 ± 27.38	605.69 ± 84.97	-
MPV(μm^3^)	6.09 ± 0.08	4.88 ± 0.13 ***	⇓

**Table 2 brainsci-13-00161-t002:** Effect of MTTA on the clinical chemistry (tCO2, Ca, PHOS, Mg, K+, Na+, CL−) in mice. The data shown here (see Materials and Methods) refer to the mean ± SEM values relating to eight animals for vehicle and eight for MTTA. Student’s *t*-test was used to determine statistical significance (*p* < 0.05) between the two groups. * *p* < 0.05; *** *p* < 0.001 versus vehicle.

Electrolyte Parameters	Vehicle	MTTA	Variations
tCO2 (mmol/L)	12.2 ± 0.27	16.8 ± 0.53 ***	⇑⇑
Ca (mg/dL)	8.9 ± 0.10	8.24 ± 0.05 ***	⇓
PHOS (mg/dL)	12.8 ± 0.89	8.30 ± 0.07 ***	⇓⇓
Mg (mg/dL)	2.1 ± 0.10	1.62 ± 0.12 ***	⇓
K^+^ (mmol/L)	6.6 ± 0.21	6.7 ± 0.20	-
Na^+^ (mmol/L)	120.0 ± 1.28	118.7 ± 0.92 *	⇓
Cl^-^ (mmol/L)	101.0 ± 0.86	103.6 ± 4.35	-

**Table 3 brainsci-13-00161-t003:** Effect of MTTA on the clinical chemistry (ALB, TP, GLOB, A/G, TB, GGT, ALT, ALP, AMY, Crea, TC, GLU, Ca, PHOS, BUN/CREA, BUN) in mice. The data shown here (see Materials and Methods) refer to the mean ± SEM values relating to eight animals for vehicle and eight for MTTA. Student’s *t*-test was used to determine statistical significance (*p* < 0.05) between the two groups. * *p* < 0.05; ** *p* < 0.01; *** *p* < 0.001 versus vehicle.

Comprehensive Parameters	Vehicle	MTTA	Variations
ALB (g/dL)	3.29 ± 0.12	3.13 ± 0.05 **	⇓
TP (g/dL)	5.22 ± 0.19	4.67 ± 0.08 ***	⇓
GLOB (g/dL)	1.58 ± 0.33	1.61 ± 0.37	-
A/G	1.19 ± 0.13	1.04 ± 0.08 *	⇓
TB (mg/L)	0.06 ± 0.00	0.14 ± 0.00 ***	⇑⇑⇑
GGT (U/L)	1.78 ± 0.00	2.11 ± 0.26 **	⇑
ALT (U/L)	109.44 ± 18.87	49.66 ± 9.11 ***	⇓⇓
ALP (U/L)	101.00 ± 3.62	46.33 ± 10.52 ***	⇓⇓
AMY (U/L)	3175 ± 18.30	2510 ± 521.07 ***	⇓
Crea (mg/L)	0.11 ± 0.02	0.66 ± 0.48 **	⇑⇑⇑
TC (mg/L)	0.00 ± 0.00	102.11 ± 19.95 ***	⇑⇑⇑
GLU (mg/L)	100.25 ± 10.67	203.92 ± 22.50 ***	⇑⇑⇑
Ca(mg/L)	8.37 ± 0.27	6.24 ± 1.15 **	⇓
PHOS (mg/L)	6.49 ± 0.53	5.74 ± 1.22	-
BUN/CREA	0.00 ± 0.00	0.00 ± 0.00	-
BUN (mg/L)	20.67 ± 0.56	21.03 ± 0.65	-

Regarding electrolyte parameters (Table 2), Ca (t = 16.70, Df = 14, *p* < 0.0001), PHOS (t = 17.43, Df = 14, *p* < 0.0001), Mg (t = 8.691, Df = 14, *p* < 0.0001) and Na+ (t = 2.333, Df = 14, *p* = 0.0351) are decreased, while tCO2 (t = 21.87, Df = 14, *p* < 0.0001) is increased. On the other hand, comprehensive parameters (Table 3) show ALB (t = 3.481, Df = 14, *p* = 0.0037), TP (t = 7.546, Df = 14, *p* < 0.0001), A/G (t = 2.779, Df = 14, *p* = 0.0148), ALT (t = 8.069, Df = 14, *p* < 0.0001), ALP (t = 13.90, Df = 14, *p* < 0.0001), AMY (t = 3.612, Df = 14, *p* = 0.0028) and Ca (t = 5.100, Df = 14, *p* = 0.0002) decrease. On the contrary, parameters TB (t = 160, Df = 14, *p* < 0.0001), GGT (t = 3.590, Df = 14, *p* = 0.0030), Crea (t = 3.238, Df = 14, *p* = 0.0060), GLU (t = 11.78, Df = 14, *p* < 0.0001) and TG (t = 160, Df = 14, *p* < 0.0001) are increased.

**Table 4 brainsci-13-00161-t004:** Effects of MTTA (0.1–30 mg/kg i.p.) on the breath rate (panel A), core temperature (panel B) and surface temperature (panel C) of mice. Breath rate values are expressed as percentage of baseline and represent the mean ± SEM of 8 mice for each treatment. Core and surface temperature are expressed as Δ °C of baseline and represent the mean ± SEM of 8 mice for each treatment. Statistical analysis was performed by two-way ANOVA followed by Bonferroni’s test for multiple comparison. * *p* < 0.05; ** *p* < 0.01; *** *p* < 0.001 versus vehicle.

Urine Parameters	Vehicle	MTTA	Variations
GLU (mg/dL)	0.00 ± 0.00	0.00 ± 0.00	-
PRO (mg/dL)	73.8 ± 12.81	141.25 ± 35.68 **	⇑⇑
BIL (mg/dL)	0.00 ± 0.00	0.00 ± 0.00	-
URO (mg/dL)	0.60 ± 0.21	5.75 ± 0.88 ***	⇑⇑⇑
pH	6.4 ± 0.16	6.6 ± 0.15 *	⇑
S.G.	1.0 ± 0.00	1.0 ± 0.00	-
BLD (mg/dL)	0.0 ± 0.01	0.1 ± 0.03 ***	⇑
KET (mg/dL)	5.6 ± 2.20	24.37 ± 8.84 ***	⇑⇑⇑
NIT	0.00 ± 0.00	0.00 ± 0.00	-
LEU(Leu/uL)	9.4 ± 4.57	134.37 ± 34.38 ***	⇑⇑⇑

## Data Availability

The data presented in this study are available on request from the first (Giorgia Corli) and corresponding author (Matteo Marti) for researchers of academic institutes who meet the criteria for access to confidential data.

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
