# Peer review of "Pharmaco-Toxicological Effects of Atypical Synthetic Cathinone Mephtetramine (MTTA) in Mice: Possible Reasons for Its Brief Appearance over NPSs Scene"

_brainsci, 2023, doi:10.3390/brainsci13020161_

Round 1

Reviewer 1 Report

Although mephtetramine (MTTA), an atypical synthetic cathinone,  was first reported in Europe in 2013, our current knowledge of its toxicological and pharmacodynamic properties is very limited.

This study aimed at examining behavioral, biochemical and pathomorphological effects induced by subchronic administration of mephtetramine in mice. In behavioral studies a battery of appropriate tests was used. In an analytical part, effects of MTTA on blood cells counts, several chemical/biochemical parameters in the blood and urine were measured. Finally, histological examinations of samples of the heart, kidney and spleen were conducted in order to reveal any injuries related to mephtetramine-treatment.

The most worrisome changes were observed in the number of blood cells (mostly decreases) and pathological findings in the heart and kidneys.

Comments:

1.      The discussion part of the MS is a little bit too speculative. I expect that at least a suggestion has been made to examine potency and efficacy of MTTA at DAT, SERT and NET. Such data would likely help to understand why several actions of MTTT differ from those of other synthetic cathinones

2.      As there are several linguistic errors in the text, it should be checked by a native English speaker. Below are examples of errors:

Line 54. Is nightclub, festivals. Should be nightclubs, music festivals.

Line 64. Is that.. Should be they.

Line 76. Should be - is may

Line 83. It is manifest that I suggest to delete this fragment.

Line 93 Should be adverse effects in humans.

Lines 110, 146. Should be were

Line 125. cm2, 2 should be in superscript

Line 143. Should be 4 ul/g body mass.

Line 434. Is Drug test. Should be Drag test

3.      I suggest to list abbreviations according to an alphabetic manner.

Author Response

We thank the Reviewer 1 for his/her evaluation of our manuscript and for helpful concerns to improve the article. In this revised version of the work, we have addressed the major concerns of the referee (changes made to the text are highlighted in yellow).

Although mephtetramine (MTTA), an atypical synthetic cathinone, was first reported in Europe in 2013, our current knowledge of its toxicological and pharmacodynamic properties is very limited.

This study aimed at examining behavioral, biochemical and pathomorphological effects induced by subchronic administration of mephtetramine in mice. In behavioral studies a battery of appropriate tests was used. In an analytical part, effects of MTTA on blood cells counts, several chemical/biochemical parameters in the blood and urine were measured. Finally, histological examinations of samples of the heart, kidney and spleen were conducted in order to reveal any injuries related to mephtetramine-treatment.

The most worrisome changes were observed in the number of blood cells (mostly decreases) and pathological findings in the heart and kidneys.

Rev1Q1: The discussion part of the MS is a little bit too speculative. I expect that at least a suggestion has been made to examine potency and efficacy of MTTA at DAT, SERT and NET. Such data would likely help to understand why several actions of MTTA differ from those of other synthetic cathinones.

AA: We thank the Reviewer 1 for pointing out this point. Given the lack of information that can clarify mechanisms underlying the atypical effects of mephtetramine, we provide further information about chemically similar compounds in the effort to avoid as much as possible speculative observations. 

Rev1Q2: As there are several linguistic errors in the text, it should be checked by a native English speaker. Below are examples of errors:

Line 54. Is nightclub, festivals. Should be nightclubs, music festivals.

Line 64. Is that.. Should be they.

Line 76. Should be - is may

Line 83. It is manifest that – I suggest to delete this fragment.

Line 93 Should be adverse effects in humans.

Lines 110, 146. Should be were

Line 125. cm2, 2 should be in superscript

Line 143. Should be 4 ul/g body mass.

Line 434. Is Drug test. Should be Drag test

AA: We thank the Reviewer 1 for pointing out these inaccuracies and provide a revised version of the manuscript with particular reference to the above-mentioned errors.

Rev1Q3: I suggest to list abbreviations according to an alphabetic manner.

AA: We thank the Reviewer 1 for the suggestion and list abbreviations according to an alphabetic manner.

Reviewer 2 Report

The presented study is extremely interesting and accurate in the conduct of the experiments.

 One missing information should be completed: starting from the fact that kidney is damaged by the administered substance, eventual damages to the liver should be reported, if any or if noone and liver observation can be added with photographs if needed.

Observations on liver histology should support clinico-toxicological data.

 Please provide an english text editing

Author Response

We thank the Reviewer 2 for his/her evaluation of our manuscript and for helpful concerns to improve the article. In this revised version of the work, we have addressed the major concerns of the referee (changes made to the text are highlighted in green).

The presented study is extremely interesting and accurate in the conduct of the experiments.

One missing information should be completed: starting from the fact that kidney is damaged by the administered substance, eventual damages to the liver should be reported, if any or if noone and liver observation can be added with photographs if needed.

Rev1Q1: Observations on liver histology should support clinico-toxicological data.

AA: We thank the Reviewer 2 for pointing out this point and add histological evaluation of liver in the specific sections of the manuscript.

Rev1Q2: Please provide an english text editing.

AA: We thank the Reviewer 2 for pointing out these inaccuracies and provide a revised version of the manuscript.